# Spatiotemporally Orchestrated Interactions between Viral and Cellular Proteins Involved in the Entry of African Swine Fever Virus

**DOI:** 10.3390/v13122495

**Published:** 2021-12-13

**Authors:** Kehui Zhang, Su Li, Sheng Liu, Shuhong Li, Liang Qu, George F. Gao, Hua-Ji Qiu

**Affiliations:** 1State Key Laboratory of Veterinary Biotechnology, National African Swine Fever Para-Reference Laboratory, Harbin Veterinary Research Institute, Chinese Academy of Agricultural Sciences, Harbin 150069, China; zkhzhangkehui@163.com (K.Z.); lisu@caas.cn (S.L.); yyqx1128lee@163.com (S.L.); tierno831143@outlook.com (L.Q.); 2CAS Key Laboratory of Pathogenic Microbiology and Immunology, Institute of Microbiology, Chinese Academy of Sciences, Beijing 100101, China; sliu520@mail.ustc.edu.cn

**Keywords:** African swine fever virus, virus entry, clathrin-mediated endocytosis, macropinocytosis, endosome pathway

## Abstract

African swine fever (ASF) is a highly contagious hemorrhagic disease in domestic pigs and wild boars with a mortality of up to 100%. The causative agent, African swine fever virus (ASFV), is a member of the *Asfarviridae* family of the nucleocytoplasmic large DNA viruses. The genome size of ASFV ranges from 170 to 194 kb, encoding more than 50 structural and 100 nonstructural proteins. ASFV virions are 260–300 nm in diameter and composed of complex multilayered structures, leading to an intricate internalization pathway to enter host cells. Currently, no commercial vaccines or antivirals are available, due to the insufficient knowledge of the viral receptor(s), the molecular events of ASFV entry into host cells, and the functions of virulence-associated genes. During the early stage of ASFV infection, the fundamental aspects of virus-host interactions, including virus internalization, intracellular transport through the endolysosomal system, and membrane fusion with endosome, are precisely regulated and orchestrated via a series of molecular events. In this review, we summarize the currently available knowledge on the pathways of ASFV entry into host cells and the functions of viral proteins involved in virus entry. Furthermore, we conclude with future perspectives and highlight areas that require further investigation. This review is expected to provide unique insights for further understanding ASFV entry and facilitate the development of vaccines and antivirals.

## 1. Introduction

African swine fever (ASF) is an acute, hemorrhagic, and severe porcine infectious disease caused by African swine fever virus (ASFV) and is a disease listed by the World Organization for Animal Health (OIE). The disease shows high mortality in domestic pigs, approaching 100%, while African wild suids coevolved with the virus have high resistance [1,2]. ASF outbreaks severely threaten the global pig industry and result in serious economic losses. Currently, the most feasible method of controlling ASF is quarantine and slaughter of infected pigs. The reservoir hosts of ASFV include wild *suids* and arthropod vectors of the *Ornithodoros* genus, both of which can act as vectors for the virus propagation [3,4]. ASFV is biologically complex in its mechanisms of entry, which limits the understanding of initial virus-host interactions.

In this review, we comprehensively compared the entry process of large DNA viruses, including ASFV, poxviruses, and herpesviruses. In addition, the molecular events of ASFV entry are summarized to provide a better understanding of ASFV pathogenesis and facilitate the design of antiviral strategies.

## 2. The Structure of ASFV Virions

ASFV, a member of the *Asfarviridae* family, is a large, enveloped virus with icosahedral morphology and an average diameter of 260–300 nm [5]. The genome of different ASFV strains ranges from 170 to 194 kb, and the number of genes of different virus strains is different, which are primarily due to the acquisition or loss of multigene family genes (MGFs) of the virus [6]. The ASFV genome encodes more than 50 structural proteins and 100 nonstructural proteins, among which the former involved the structural components of the virions, while the latter is associated with viral genome replication, transcription, translation, and mRNA modification [7,8].

ASFV replicates in porcine alveolar macrophages (PAMs). Viral DNA replication and virus morphogenesis occur in specific cytoplasmic sites close to the nucleus, which are called viral factories [9,10]. Furthermore, the nucleus is also a site of viral DNA synthesis during the early stage of viral infection [11,12]. Recently, our group and another team dissected the structure of the ASFV virions, which is comprised of a genome containing nucleoids, a core shell, an inner lipoprotein envelope, a capsid, and an outer envelope (Figure 1) [5,13]. Moreover, another group revealed a six-layered structure of ASFV virions, including nucleoids, core shell, inner capsid, inner lipoprotein envelope, outer icosahedral protein capsid, and lipoprotein membranes [14]. To date, CD2v is the only identified viral protein located in the outer envelope, while seven viral proteins (p12, p17, p54, p22, pH108R, pE199L, and pE248R) are localized at the inner envelope. The icosahedral capsid consists of the major capsid protein (MCP) p72, the minor capsid protein p49, and the protein pE120R. More recently, we also identified the H240R protein as a novel capsid protein interacting with the MCP p72 during the morphogenesis of ASFV [15]. The core shell is composed of the mature products p150, p37, p34, p14, and p5 derived from the polyprotein pp220 and p35, p15, and p8 derived from pp62 (Figure 1). The minor capsid proteins form a hexagonal network below the outer capsid shell, functioning as stabilizers by ‘‘gluing’’ neighboring capsomers together [5].

The ASFV virions contain not only viral proteins but also host molecules, which are acquired during virion assembly and budding. The virion assembly is characterized by a highly coordinated process involved in the expression of different viral genes [14,16]. The outer capsid and inner core shell above and beneath the viral membrane are gradually assembled, resulting in the enclosure of the genomic material, giving rise to the intracellular icosahedral mature virus [17]. The mature virus is formed by the progressive assembly of the outer capsid, move to the cell surface by microtubule-mediated transport, and are released from cells by budding at the plasma membrane, which does not follow an icosahedral symmetry [18,19]. Interestingly, ASFV virions with the *H240R* gene-deletion can be released from PAMs by budding through the plasma membrane but produce aberrant virions with tubules and bilobular structures [15]. Using mass spectrometry (MS) analysis, 68 viral proteins and 21 host proteins (including annexin, integrin, and β-actin) were identified in ASFV particles. However, only a few viral and cellular proteins in the localization of ASFV particles have been elucidated to date [8], indicating the complexity of the ASFV virions.

## 3. Overview of the Entry Process of Large DNA Viruses

ASFV, herpesviruses, and poxviruses are large DNA viruses, which can hijack cellular factors for virus entry with some similar pathways. Herpesviruses infect almost all vertebrates and several invertebrates [20]. The envelope proteins of herpesviruses mediate virus attachment and fusion, and the attachment of herpesviruses on the cellular surface requires heparan sulfate proteoglycans (HSPGs) [21]. Notably, two herpesviruses, herpes simplex virus 1 (HSV-1) and HSV-2, recognize different domains of HSPGs [22]. Furthermore, the envelope glycoprotein C (gC) of HSV recognizes HSPGs for virus attachment, whereas gC is not essential for viral entry. Moreover, the envelope gB can also mediate virus attachment through binding HSPGs in the absence of gC [23,24]. Besides, the binding of the envelope glycoprotein gD to cell surface receptor(s) is a key step for entry of HSV-1, HSV-2, and other *alphaherpesviruses*. It has been reported that HSV-1 entry requires the interaction of gD and the herpesvirus entry mediator (HVEM) or nectin1 receptor, and the fusion with cell membrane through the viral fusion complex (gB, gH, and gL) [25,26].

Poxviruses can infect both invertebrates and vertebrates, and the virus entry is the process by which the nucleoprotein core of the virus enters the cytoplasm. Mature vaccinia virus (VACV), a member of poxviruses, can enter the plasma membrane at neutral pH or through a low pH-dependent pathway. After entry into host cells, the viral envelope proteins fuse with an endocytic vesicle, followed by the release of the viral core into the cytoplasm [27,28]. The VACV genome encodes several proteins involved in viral adsorption and entry. Four viral proteins (D8, A26, A27, and H3) mediate the attachment of mature VACV, whereas 11 proteins (A16, A21, A28, G3, G9, H2, J5, L5, O3, F9, and L1) form the viral entry/fusion complex (EFC) and exert the membrane fusion function [29].

The entry of ASFV can be summarized by the following aspects: (1) ASFV binding to the cell surface via an unknown receptor(s) or nonselective uptake of extracellular particles by macropinocytosis; (2) activation of cellular signaling pathways; (3) ASFV entry into host cells through clathrin-mediated endocytosis (CME) and actin-driven macropinocytosis, endocytosis and transport to endosomes in a stepwise pH-dependent process; and (4) fusion of inner envelope proteins with membranes of endosomes and the delivery of naked cores into the cytosol. It has been reported that viral membrane fusion is dependent on the integrity of cholesterol efflux from the endosomes since the virions would be retained inside the endosomes without cholesterol efflux [30]. The viral proteins pE248R and pE199L of ASFV involved in the membrane fusion process share sequence similarities with the VACV proteins L1, G9, A16, and J5 [31,32]. Therefore, ASFV virions may enter host cells through similar pathways as VACV.

## 4. Molecular Events of ASFV Entry

Entry into host cells is an initial event of viral infection [33]. The entry of ASFV into host cells is a process involving precise regulation and mutual coordination, which follows a specific spatiotemporal order [34]. Previous studies have shown that ASFV enters host cells through CME and actin-driven macropinocytosis (Figure 2) [35,36,37,38]. After binding to host cells, the virions stimulate the uptake of dextran by the cells, which is followed by activation of the epidermal growth factor receptor (EGFR) and PI3K-Akt signaling pathways to facilitate ASFV entry [39]. Furthermore, the successful internalization of virions depends on the action of dynamin, the efflux of cholesterol at the membrane, and the rearrangement of actin [30,40]. In addition, endocytosis is also a temperature- and energy-dependent process, since the internalization is completely blocked in the presence of metabolic inhibitors or upon incubation at 4 °C [41].

### 4.1. ASFV Entry Is a Dynamin-Dependent and CME Process

CME is a process of receptor-dependent internalization of virus particles, characterized by the formation of clathrin-coated pits (CCPs) that bud into the cytoplasm to form clathrin-coated vesicles (CCVs) [35,42]. During the infection, ASFV can enter host cells via CME, which requires cholesterol flux [30]. Moreover, the GTPase dynamin plays a critical role in CCPs budding into the cytoplasm. It has been demonstrated that PAMs treated with the dynamin inhibitor dynasore (DYN) exhibit a remarkable reduction of ASFV uptake [40], indicating that dynamin-mediated membrane scission is involved in the CME process of ASFV (Figure 2). Upon endocytosis, incoming ASFV virions move along the endolysosomal pathway to detach the viral envelope and capsid.

### 4.2. ASFV Enters Host Cells by Macropinocytosis

Macropinocytosis is a non-selective uptake process of extracellular particles via actin-dependent evaginations of the plasma membrane to form large uncoated vesicles of 0.5–10 μm [37], which is another important endocytic pathway that is hijacked by various viruses for entry into host cells [43]. Endocytosis is characterized by the formation of ruffles or blebs in the plasma membrane induced by the activation of kinases and Rho GTPases [44]. It has been reported that macropinocytosis is not induced by ASFV particles, but a constitutive process that occurs in macrophages [40]. During the ASFV entry, actin reorganization and blebbing in macropinocytosis can facilitate the CME process. The entry process via macropinocytosis and CME may be coordinated to facilitate ASFV entry. Upon internalization, the virions are transported from early endosomes (EEs) to late endosomes (LEs). Subsequently, the virus undergoes outer envelope disruption and capsid disassembly, exposing the inner envelope and then fusing with the endosomal membrane to release the naked genome-containing core into the cytoplasm. Furthermore, a fraction of disrupted particles reaches lysosomes, and then undergo membrane fusion or be further degraded [33,39,45,46,47]. Thus, lysosomal hydrolases may facilitate virus disruption or uncoating.

Moreover, ASFV also infects vascular endothelial cells or epithelial cells with low efficiency in vitro. Additionally, several cell-adapted ASFV strains can efficiently replicate in Vero cells [48]. Interestingly, we have recently demonstrated that ASFV can adapt to HEK293T cells after serial passages, and the adapted ASFV can replicate efficiently in both HEK293T and Vero cells [49]. Thus, ASFV exploits multiple entry strategies to adapt to the changing infection conditions in different cells.

### 4.3. Intracellular Transport of ASFV

After internalization into host cells, the ASFV particles in the endosome move along the endosome-lysosome pathway. The sequential maturation of EEs to LEs depends on endosomal membrane signaling modulated by both proteins and lipids (cholesterol) [50,51]. The mature endosome is characterized by the expression of CD63 and Rab7 GTPase (Rab7) [40,44]. Rab7 controls the transport and fusion of LEs. The endosomal maturation pathway orchestrated by Rab proteins and phosphoinositides plays a critical role during the early stages of ASFV infection [44]. The colocalization of EEA1 or Rab5 with the virions can be detected from 5 to 30 min postinfection (mpi), while the interactions of the envelope protein (p12) or core (p150) with CD63, Rab7, or Lamp1 can be observed from 30 to 90 mpi using laser confocal microscopy [40] (Figure 2), indicating that CD63, Lamp1, and GTPases (Rab5 and Rab7) are important for ASFV intracellular transport.

### 4.4. ASFV Uncoating and Fusion

The decomposition process is gradually driven by the acidic pH in the endosome, which is required for the ASFV life cycle. It has been shown that PAMs or Vero cells treated with bafilomycin A1 (an inhibitor of endosome acidification) prior to ASFV infection significantly prevented virus disassembly [52], suggesting that endosome acidification is needed during the virus uncoating.

Most ASFV virions inside EEs appear as nearly intact structures of mature viral particles [30]. However, low pH in the EEs triggers conformational changes in the capsid structure, resulting in the dissociation of their components and the detachment of the outer membrane [40]. Therefore, most virions (>85%) lose the capsid proteins in LEs, while a significant proportion of virions (>50%) lack the outer envelope proteins [4]. After the decomposition process, the viral inner envelope pE199L and pE248R proteins fuse with the endosomal membrane of LEs, releasing the naked core into the cytosol (Figure 2). The mature core is separated from the inner envelope upon hydrolysis of the pp220 protein. A recent study has tracked the entry process of virions using confocal microscopy, the occurrence of either p12^+^p150^−^ or p12^+^p150^+^ particles in cell staining with anti-p12 and anti-p150 antibodies indicates the cell-bound extracellular virions or endocytosed particles, respectively, whereas the presence of p12^−^p150^+^ punctate structures is an indicator of core penetration [31]. Since pE199L and pE248R participate in ASFV membrane fusion, the formation of a fusion complex between pE199L and pE248R remains to be elucidated. Moreover, cholesterol is a key component during the fusion of the virus with the endosomal membrane, and the transport of cholesterol in the endosomal membrane is necessary for the ASFV core release and penetration into the cytoplasm [30,37].

## 5. Viral Proteins and Host Factors Involved in ASFV Entry

### 5.1. Structures and Functions of Viral Proteins Involved in ASFV Entry

Although several studies have predicted that numerous ASFV proteins are involved in viral entry, only a few viral proteins, including p12, p54, p30, CD2v, pE248R, and pE199L, were confirmed by experimental evidence [31,32,53,54] (Table 1). Therefore, the structures and functions of these viral proteins are discussed as follows.

p12, encoded by the *O16R* gene with an approximate molecular mass of 6.9 kDa, is synthesized during the late stage of viral infection. The p12 protein is a structural protein located in the inner envelope of ASFV [8]. The recombinant p12 protein was shown to be able to block the specific binding of viral particles to susceptible cells and inhibit virus infection [54]. However, an anti-p12 antibody could not inhibit the binding of the virus to host cells or neutralize the virus [55]. Although these previous reports show conflicting evidence for the role of p12 in ASFV entry, further investigation is needed to determine the exact role of p12.

p54, encoded by the *E183L* gene with a relative molecular weight of 19.9 kDa, is the primary structural protein involved in virus transport. p54 is a type I transmembrane protein that is located on the inner envelope of the virus [56]. The p54 protein undergoes homodimerization via the formation of disulfide bonds between the conserved cysteine residues. A previous study has shown that p54 engages in the attachment of ASFV [57]. However, the p54 protein is located on the viral inner envelope, and further investigation is needed to testify that the p54 protein is involved in virus attachment or internalization.

p30, encoded by the *CP204L* gene, has a relative molecular weight of 23.6 kDa [58]. The p30 protein is an immediate early-expressed viral protein that is involved in virus internalization. It is generally accepted that the proteins involved in the viral entry are of great importance in the development of ASF vaccines [59,60]. The p54 and p30 proteins have also been demonstrated to serve as important antigenic structural proteins [61]. Through ubiquitin modification of a eukaryotic expression vector harboring ASFV hemagglutinin (sHA), the p54 and p30 proteins were shown to provide partial protection against the ASFV challenge.

CD2v, encoded by the *EP402R* gene, has a relative molecular weight of approximately 46.5 kDa and is located in the outer membrane of the virus. CD2v is a glycoprotein that resembles the T lymphocyte surface adhesion molecule CD2, which contains a signal peptide, an extracellular N-terminal domain composed of two immunoglobulin-like domains, and a transmembrane region, while the cytosolic C-terminal domain shares no obvious amino acid sequence homology with the cellular CD2 cytoplasmic domain (Figure 3). The protein is located in the Golgi network around viral factories in ASFV-infected cells. CD2v exhibits immunosuppressive activity by inhibiting the proliferation and functions of lymphocytes [62,63,64]. It has been shown that the CD2v interacts with the SH3P7/mabp1 protein through binding to its SH3 domain [65]. SH3P7/mabp1 is an actin-binding protein that contains an actin depolymerization factor homology (ADF-H) domain, functions as protein transportation through the Golgi network, participates in endocytosis mediated by grid proteins, forms membrane folds, and regulates signaling pathways [65,66,67]. The interaction between CD2v and SH3P7/mabp1 is possibly involved in virus transport in Golgi or endosome pathways. In addition, CD2v enables the adsorption of pig red blood cells onto ASFV-infected cells and promotes virus dissemination in pigs [62]. As an outer envelope protein, it may mediate the ASFV attachment to target cells, which requires validation.

pE248R, a transmembrane protein encoded by the *E248R* gene with a relative molecular weight of 27.7 kDa, is located in the inner envelope of ASFV virions. pE248R is required for the postentry step in ASFV infection [32,40] and shares amino acid sequence similarity with the VACV L1 protein, a component of the poxviral EFC that is necessary for membrane fusion and/or core penetration [29,40,68]. The pE248R protein is not required for viral disassembly but is required for fusion and core delivery into the cytosol. Upon deletion of the *E248R* gene, the morphogenesis and release of the virus can proceed normally, but the core of the virus accumulates in the lysosome-like structure and cannot successfully enter the cytoplasm, resulting in noninfectious viral progeny due to postentry blockage [32,40]. The results indicate that the pE248R is required for ASFV infectivity and is involved in an early postentry event.

pE199L (also called J18L) is another type I transmembrane protein that localizes in the inner viral envelope. The pE199L protein has a relative molecular mass of 22.7 kDa and forms cytosolic intramolecular disulfide bonds. Its amino acid sequence is highly conserved among different ASFV strains. It contains a cysteine-rich N-terminal region, a transmembrane domain, and a C-terminal tail. pE199L shares similarities with the EFC A16, G9, and J5 proteins of poxviruses [31,69]. pE199L-defective virions cannot complete membrane fusion and release the core into the cytoplasm, and the defective virions are approximately 100-fold less infectious than the parental viruses. pE199L and pE248R are not necessary for viral assembly and release but are required for viral membrane fusion and core delivery into the cytoplasm, and these proteins are essential for the progeny virus infection [31]. A recent study has demonstrated that pE199L induces cell death via the mitochondrial-dependent apoptosis pathway [70]. In addition, the MCP p72 and the inner envelope proteins, including p17, p22 and p54, are the structural proteins of ASFV that are involved in morphogenesis [71,72,73,74,75,76], but the engagement of these proteins in ASFV entry remains to be elucidated. Altogether, the viral proteins associated with virus entry are of great importance in the development of subunit vaccines against ASF in the future.

**Table 1 viruses-13-02495-t001:** The functional roles of ASFV structural proteins involved in virus entry.

ORFs	Viral Proteins	Localization/Functions	References
* EP402R *	CD2v	Outer envelope/ Hemadsorption	[64]
* CP204L *	p30	Involved in viral entry	[59]
* O61R *	p12	Inner envelope/Attachment protein	[8,54]
* D117L *	p17	Inner envelope/Morphogenesis	[71]
* B646L *	p72	Capsid/Morphogenesis	[72,73]
* E183L *	p54	Inner envelope/Morphogenesis	[74,75]
* E248R *	pE248R	Inner envelope/Entry and membrane fusion	[32]
* E199L *	pE199L	Inner envelope/Entry and penetration	[31]
* KP177R *	p22	Inner envelope	[76]

### 5.2. Host Cellular Factors Required for ASFV Entry

To date, several host cellular factors, including actin, EGFR, dynamin, and clathrin, have been identified to be involved in ASFV entry (Table 2). The dynamin and clathrin proteins are involved in the CME process. Actin-cytoskeleton rearrangements, activation of the EGFR and phosphoinositide 3-kinases (PI3Ks) signaling pathways, and the involvement of Na^+^/H^+^ exchangers could facilitate a macropinocytosis-mediated endocytic process for ASFV entry [39].

The binding of a virus and cellular receptor(s) is the initial step of viral infection. The scavenger receptor CD163 is involved in macrophage maturation. It has been reported that treatment with various anti-CD163 monoclonal antibodies can block ASFV infection in PAMs. Therefore, CD163 was considered to be a functional receptor for ASFV [77]. However, a recent study has demonstrated that gene-edited pigs lacking the *CD163* gene are fully susceptible to ASFV infection [78]. Thus, CD163 may not be the key determinant of ASFV infection in pigs.

EGFR is associated with actin rearrangement and the activation of the Rho family GTPases [79]. It has been shown that the activation of the Rho family GTPases can trigger macropinocytosis. The EGFR-specific inhibitor 324674 dramatically inhibited ASFV uptake by Vero cells [39].

PI3Ks play essential roles in growth factor signal transduction, regulation of endocytosis, and vesicular membrane trafficking. Akt is a major downstream effector of the PI3K pathway. ASFV infection induced activation of the PI3K-Akt axes, whereas treatment with the PI3K inhibitor LY294002 markedly inhibited viral entry into Vero cells [39,80].

The Rac family small GTPase 1 (Rac1) regulates cell growth, cytoskeletal reorganization, and the activation of protein kinases. It has been demonstrated that Rac1 controls macropinocytosis and regulates actin cytoskeleton dynamics [81,82]. ASFV infection triggers Rac1 accumulation in ruffling areas, whereas the Rac1 inhibitor NSC23766 significantly decreases ASFV entry into Vero cells [39].

P21-activated kinase 1 (PAK1) is involved in the regulation of cytoskeletal dynamics and is required for macropinocytosis [83,84]. PAK1 activation induced by ASFV infection remains controversial. It was suggested that PAK1 activation played a key role in ASFV infection [39], whereas another study revealed that purified virions cannot induce the activation of PAK1 [40]. The difference is likely to result from the use of clarified supernatants or purified virions in these studies.

Microtubules participate in the process of endosome maturation. During the early stages of ASFV replication, intracellular transport of virus from the plasma membrane to the perinuclear region relies on microtubule dynamics and the interaction with the virus. Colchicine and its analog MTC, which are the inhibitors of tubulin, efficiently inhibit viral migration [85,86]. Therefore, microtubules are required for the transport of the virus to the perinuclear area.

Cholesterol is required to ensure membrane fluidity for CME, which is involved in the ASFV life cycle. Cholesterol efflux from endosomes is required for ASFV fusion. The cholesterol efflux is governed by the Niemann-Pick type C 1 (NPC1) and NPC2 proteins [87]. Disruption of the cholesterol efflux by U18666A severely impaired ASFV entry. In addition, the reduction of cholesterol by statins also markedly inhibited ASFV infection [44]. A recent study showed that the LEs integral membrane protein NPC1 interacted with the inner envelope proteins pE248R and pE199L of ASFV, while NPC1 knockout reduced viral infectivity and viral replication (Preprint) [88]. Collectively, these host cellular proteins may serve as novel targets for the development of antivirals that block the initial stages of ASFV infection.

**Table 2 viruses-13-02495-t002:** The cellular factors involved in ASFV entry.

Cellular Proteins	Functions in ASFV Entry	References
Actin	Formation and trafficking of macropinosomes	[39,44]
Myosin II	Blebbing and macropinocytosis	[39]
EGFR	Actin rearrangement and activation of Rho family GTPases	[34]
Dynamin	Involved in virus uptake via clathrin-mediated endocytosis	[39,89]
Clathrin	Assembly of coated pits and clathrin-mediated endocytosis	[39,89]
PAK1	Involved in the regulation of cytoskeleton dynamics and is required during all stag e s of macropinocytosis	[83,84]
PI3K	Involved in macropinocytosis-mediated ASFV entry	[39,80]
Rac1	Modulates actin cytoskeleton dynamics and controls macropinocytosis	[39,81]
Microtubules	Transport of the virus to perinuclear area	[86]
Cholesterol	Required for ASFV entry into the cytosol	[30,41]
Rab7 GTPase	A key regulator of late endosome maturation	[39,51]
NPC1	Facilitate ASFV membrane fusion and core penetration	(Preprint) [88]
NPC2	NPC2 knockdown reduces ASFV infection	(Preprint) [88]

## 6. Concluding Remarks and Prospects

ASF outbreaks have resulted in huge economic losses to the pig industry worldwide. ASF emerged in China in 2018 and quickly spread throughout the country [90]. To facilitate the development of novel antiviral strategies, many efforts have been devoted to elucidating the functions of ASFV proteins and strategies of immunoevasion in recent years [15,49,91,92,93,94,95,96]. It is generally accepted that ASFV entry is a continuous and dynamic process orchestrated by the interactions between multiple viral and cellular proteins, which are involved in virus binding, internalization, transportation, uncoating, and membrane fusion. Notably, the application of new technologies has recently enriched the understanding of ASFV entry. Emerging single-virus tracking technology has enhanced the in-depth knowledge of virus entry into living cells [97,98,99,100]. The molecular events of virus entry can be addressed at spatial and temporal resolutions in living cells. However, only a few viral proteins and the interacting cellular factors during ASFV entry have been identified so far. Therefore, several important issues related to ASFV entry need to be further studied, including (1) the identification of the cell receptor(s) hijacked by the virus for attachment and entry; (2) the application of single-virus tracking in the entry process of living cells; (3) the elucidation of the structure and function of “EFC”; and (4) the spatiotemporal regulation of viral proteins involved in the endocytosis and fusion pathways. Increasing knowledge on ASFV entry will contribute to the comprehensive understanding of the complex interplays between ASFV and the host and provide the molecular targets for the development of vaccines or antivirals against ASF.

## Figures and Tables

**Figure 1 viruses-13-02495-f001:**
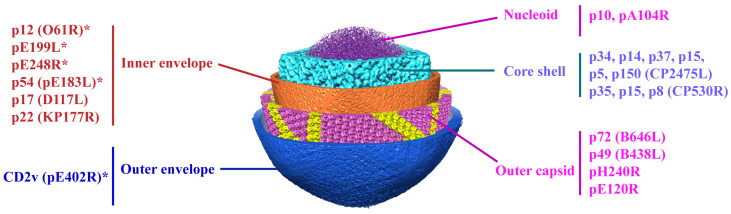
The subviral localization of viral proteins among the five-layered structure of the ASFV particle. The viral proteins involved in virus entry are mainly localized in the layers of the inner envelope, capsid, and outer envelope of ASFV. * Viral proteins involved in ASFV entry.

**Figure 2 viruses-13-02495-f002:**
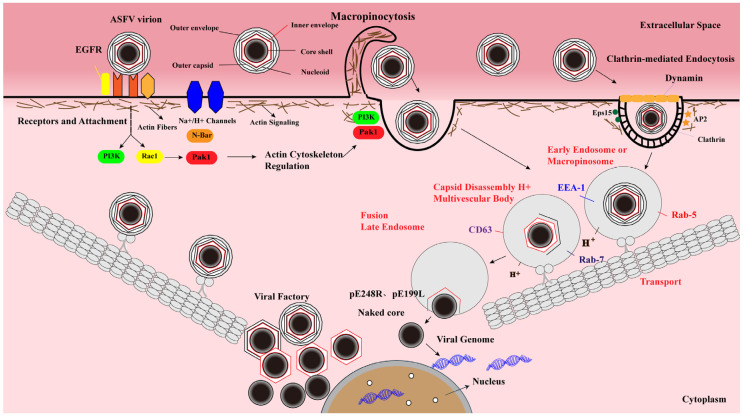
ASFV enters host cells through a complex process involving dynamin- and clathrin-mediated endocytosis (CME) and micropinocytosis. After ASFV entry into the cells via CME and micropinocytosis, the virions are transported from early endosomes to late endosomes along the cytoskeleton, where they undergo an uncoating process characterized by pH-dependent capsid disassembly and disruption of the outer viral envelope. The inner envelope proteins pE248R and pE199L are fused with the endosomal membrane to deliver the genome-containing “naked” cores into the cytosol. Viral genomic DNA replication and virion assembly take place in the viral factories. The newly synthesized virions then move toward the plasma membrane, where they acquire the outer envelope by budding.

**Figure 3 viruses-13-02495-f003:**
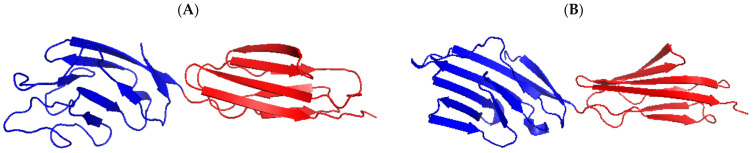
Predicted three-dimensional structure of the CD2v extracellular domain. (**A**) Homologous modeling analysis of the CD2v protein was performed using the software PyMOL 1.7 according to the structure of the human T cell adhesion molecule CD2 protein (**B**). The crystal structure of human T cell adhesion molecule CD2 protein (SMTL ID:1hnf.1). Immunoglobulin-like domains 1 and 2 were shown in blue and red, respectively.

## Data Availability

All data generated or analyzed during this study are included in this published article.

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
