# Peer review of "Spatiotemporally Orchestrated Interactions between Viral and Cellular Proteins Involved in the Entry of African Swine Fever Virus"

_viruses, 2021, doi:10.3390/v13122495_

Round 1

Reviewer 1 Report

This paper presents a timely review of the known mechanisms regulating viral entry of the highly infectious haemorrhagic African swine fever virus (ASFV) into its target cells. ASFV has a complex genome for a virus and relatively few detailed molecular-level investigations have been presented. This review gives us the status of the current knowledge on ASFV entry to its target cells. Some discussion is presented on host cell genes required for ASFV infection. Developing an understanding of the molecular basis of cell entry of ASFV may allow the development of effective therapeutic strategies against this deadly virus. This review points to the paucity of data available for this complex and economically important virus, and more generally the lack of detailed understanding of viral entry and action.

Paragraph1, section 3, lines 84-95: This paragraph is a little confusing. The abbreviations for the envelope glycoproteins gC, gB, gD, nor their importance are explained in the text.

Some editing would help this manuscript. Many of the sentences have multiple sub-clauses making them more complicated than necessary.

Minor points

Line 31: the high mortality of ASFV really only pertains to domesticated swine, wild African suids have co-evolved with the virus and have some immunity

Line 33: viable rather than liable

Line 51: ...in the “viral factory”…This sentence has too many sub-clauses and should be split, I would suggest a new sentence at: The nucleus is also…

Line 163 &178: should LEs be LYs? Otherwise define LEs

Line186-187: The meaning of this sentence is not clear, is there something missing? ..but whether the generation of a fusion complex occurs remains…?

Line 225: there should be a reference inserted after ‘…vaccines’ to substantiate the statement.

Reviewer 2 Report

The paper presents additional review on African swine fever virus genetic features which were previously published in many reputable and well-written papers. I fell the repetition of additional data will not bring any advantage to the state of knowledge on ASFV. 

Example

Galindo I, Alonso C. African Swine Fever Virus: A Review. Viruses. 2017;9(5):103. Published 2017 May 10. doi:10.3390/v9050103
